# Changes in the Velocity of Blood in the Portal Vein in Mild Acute Pancreatitis—A Preliminary Clinical Study

**DOI:** 10.3390/medicina55050211

**Published:** 2019-05-26

**Authors:** Artautas Mickevičius, Jonas Valantinas, Juozas Stanaitis, Tomas Jucaitis, Laura Mašalaitė

**Affiliations:** 1Vilnius University Faculty of Medicine, Clinic of Gastroenterology, Nephro-Urology and Surgery, Biomedical Sciences, Vilnius LT-03101, Lithuania; jonas.valantinas@santa.lt (J.V.); laura.masalaite@santa.lt (L.M.); 2Centre of Hepatology, Gastroenterology and Dietetics, Vilnius University Hospital Santaros Clinics, Vilnius LT-08661, Lithuania; juozas.stanaitis@santa.lt (J.S.); tomas.jucaitis@santa.lt (T.J.)

**Keywords:** acute pancreatitis, portal vein thrombosis, portal venous velocity, color Doppler ultrasound

## Abstract

*Background and objective*: Portal vein thrombosis is associated with a decrease in the main blood velocity in this vessel. While most studies examine etiological factors of portal vein thrombosis after its occurrence, we aimed to evaluate portal vessels and assess whether mild acute pancreatitis affects blood flow in the portal vein and increases the risk of thrombosis. *Materials and methods*: This prospective single centered follow-up study enrolled 66 adult participants. Fifty of them were diagnosed with mild acute pancreatitis based on the Revised Atlanta classification, and 16 healthy participants formed the control group. All participants were examined three times. The first examination was carried out at the beginning of the disease and the next two at three-month intervals. Blood samples were taken and color Doppler ultrasound performed the first time, whereas ultrasound alone was performed during the second and third visits. Mean and maximal blood velocities and resistivity index in the main portal vein and its left and right branches were evaluated. *Results*: Mean velocity of the blood flow in the main portal vein and its right and left branches was not significantly different from healthy individuals during the acute pancreatitis phase: 23.1 ± 8.5 cm/s vs. 24.5 ± 8.2 cm/s (*p* = 0.827); 16.4 ± 7.9 cm/s vs. 16.4 ± 8.1 cm/s (*p* = 1.000); and 8 ± 3.4 cm/s vs. 7.4 ± 2.5 cm/s (*p* = 0.826), respectively. The same was observed when comparing the maximal blood flow velocity: 67.9 ± 29 cm/s vs. 67.5 ± 21 cm/s (*p* > 0.05); 45.4 ± 27 cm/s vs. 44 ± 23.8 cm/s (*p* = 0.853); and 22.2 ± 9.8 cm/s vs. 20 ± 7.3 cm/s (*p* = 0.926), respectively. Changes in venous blood velocities were not significant during the follow-up period in separate study groups. *Conclusions*: Portal blood flow velocities do not change during mild acute pancreatitis in the inflammatory and postinflammatory periods. This observation suggests that mild acute pancreatitis does not increase the risk of portal vein thrombosis.

## 1. Introduction

Acute pancreatitis (AP) is one of the most common gastrointestinal disorders with incidence rates increasing in recent years [1,2]. Worldwide, the reported annual incidence of AP ranges from 36 to 78 per 100,000 population [3,4]; however, approximately 80–90% of cases are of a benign clinical course, and only a minority of patients develop severe AP [5]. Alcohol and gallstone disease are known to be two of the most common causes for the development of AP [6]. One third of AP cases are of metabolic (alcohol induced) origin. High alcohol intake and duration of alcohol consumption are both associated with an increased risk of pancreatitis [6,7]; however, the toxic effect of alcohol is not completely clear. Studies suggest that ethanol induces oxidative stress and stimulates secretion of pancreatic digestive enzyme, and metabolites of ethanol, like fatty-acid ethyl esters, induce pancreatic injury, thereby causing inflammation of the pancreas [7]. Although biliary origin is the leading cause in the modern world, the mechanism by which the passage of gallstones induces pancreatitis is unknown. Two concepts prevail: reflux of infected bile into the pancreas activates a cascade of proteolytic enzymes; and obstruction of pancreatic duct causes acinar disruption due to raised pressure [8]. Other etiological factors of AP, such as trauma, medication, and viral or autoimmune disorder, are less common.

The Revised Atlanta Classification classifies AP into mild, moderately severe, and severe. This is based on the presence and duration (less or more than 48 h) of organ failure, and local and systemic complications [9]. Mild acute pancreatitis is a self-limiting disease with no signs of local or systemic complications and comprises up to 80% of all AP cases. Moderately severe disease presents with local or systemic complications, or organ failure lasting less than 48 h. When organ failure last longer, AP is classified as severe [10].

Doppler ultrasound (US) in a noninvasive, relatively cheap, and easily accessible imaging modality that helps in diagnosing portal vein thrombosis and assessing portal vessels. It is the current clinical standard for identifying hemodynamic changes in portal venous flow in patients with hepatic disease [11]. Researchers have demonstrated a significant correlation between main blood flow volume and mean blood velocity in the portal vein, and the liver cirrhosis stage [12,13]; however, a decrease in the mean blood flow velocity in the main portal vein may be also a sign of portal vein thrombosis [14].

Portal vein thrombosis, as well as thrombosis in general, is a multifactorial condition caused by the interaction between three factors: inherited or acquired prothrombotic disorders, other thrombophilic factors, and local factors [11]. Local factors seem to work as triggers for acute portal vein thrombosis in the case of latent prothrombotic state (thrombophilia) [11]. Local factors (precipitating thrombosis) include acute inflammation of pancreas, or chronic pancreatitis in the case of acute events or formation of a pseudocyst [15]. Development of inflammation leads to activation of coagulation. The inflammatory process leads to microvascular thrombosis by activating the coagulation system, reducing the activity of natural anticoagulants, and disturbing functioning of the fibrinolysis system. In early pathogenesis of AP, microvascular abnormalities are very important. While in mild AP, pancreatic capillary blood flow increases; severe AP is associated with early impairment of pancreatic blood flow—complete capillary stasis is observed in almost 40% of pancreatic capillaries [16]. While most studies focus on examining etiological factors of portal vein thrombosis after its occurrence, we aimed to evaluate portal vessels before the thrombotic event and assess whether mild acute pancreatitis, comprising the majority of pancreatitides, affects blood flow in the portal vein and its branches and increases the risk of thrombosis.

In this prospective follow-up study, we aimed to:Evaluate changes in the velocity of blood flow in the portal vein at the time of diagnosis of acute pancreatitis.Evaluate changes in the velocity of blood flow in the portal vein during the follow-up period after the acute pancreatitis treatment.Evaluate the risk of portal vein thrombosis in the case of mild acute pancreatitis in the absence of other prothrombotic factors.

## 2. Materials and Methods

Ethics approval and consent to participate. The local Lithuanian Bioethics Committee approved protocol of this study (No. 158200-15-774-291, 2015-05-03), and each individual signed an individual informed consent form prior to participation. The ethical principles of the Declaration of Helsinki for medical research involving human subjects were fulfilled [17]. There has been no significant financial support for this work that could have influenced its outcome.

Patients. The prospective single-centered preliminary follow-up study was conducted between May 2015 and April 2018 in Vilnius University Hospital Santaros Clinics, in the Centre of Hepatology, Gastroenterology and Dietetics. Sixty-six adult participants were enrolled for the present study with age ranging from 23 to 69 years. Fifty patients diagnosed with mild acute pancreatitis were included in the study (acute pancreatitis group) at the time of diagnosis. Based on the International Association of Pancreatology and American Pancreatic Association guidelines, AP was diagnosed when two of three criteria were met: clinical (upper abdominal pain), laboratory (serum amylase or lipase three times higher than the normal limit), and/or imaging (specific changes in computed tomography, magnetic resonance, or ultrasonography) [18]. All cases of AP were of a mild form based on the Revised Atlanta Classification [9]. Patients with mild AP were hospitalized after being admitted to the emergency room and treated in the in-patient department for 5 to 7 days. The treatment included intravenous fluid therapy (5 mL/kg/day intravenous fluid therapy (Ringer’s and/or 0.9% sodium chloride solution) in the first day of hospitalization with the infusion speed of 200–300 mL per hour, reducing to 1.5 L per day at the end of hospitalization), enzyme replacement therapy, analgesics (nonsteroidal anti-inflammatory drugs), and diet. Anticoagulants are not a part of the AP treatment algorithm and were not prescribed for patients. During the follow-up period, after being discharged from the hospital, 21 patients did not come to the second or third visit; therefore, they were excluded from the further study (Figure 1). Sixteen healthy volunteers with no signs or symptoms of acute or chronic pancreatitis, and no history of acute pancreatic inflammation formed the control group. Individuals with liver disease and/or its complications, myeloproliferative disorders, oncologic diseases, heart and circulatory diseases, and coagulation disorders were excluded from the study on the basis of medical history, clinical and laboratory findings, and imaging of the liver, bile ducts, pancreas, and other abdominal organs. 

Methods. All participants were examined three times over a follow-up period of six months. The first examination was carried out at the beginning of the disease and the next two at three-month intervals. Blood samples were taken and color Doppler ultrasound was performed the first time; only abdominal ultrasound was performed during the second and third visits. 

All patients and control subjects were asked to fast for at least 6 h before blood sampling and US examination. 2.7 mL and 5 mL venous blood samples were collected in sodium citrate and EDTA tubes by a nurse during venipuncture procedure in the blood-sampling room. In case of mild acute pancreatitis, sampling was performed within 24 h of admission to the hospital. Blood samples of healthy volunteers were collected in the morning before the US examination. Samples were analyzed in the Centre of Laboratory Medicine in Vilnius University Hospital Santaros Clinics. Complete blood count (Beckman Coulter Diagnostics analyzer, Atlanta, USA), coagulation markers (activated partial prothrombin time (APPT), international normalized ratio (INR), prothrombin time, fibrinogen, D-dimer) (Stago group analyzer, Asnieres-sur-Seine, France), and biochemical markers (electrolytes, C-reactive protein, lipase, *p*-amylase, and lactate dehydrogenase) (Abbott analyzer, Texas, USA) were performed for each study participant. Blood tests were not repeated during the follow-up period. At the same day as blood sampling, the first abdominal color Doppler ultrasound was performed using a Toshiba diagnostic ultrasound system, SSA-790A ultrasound machine (Toshiba medical systems corporation, Otawara, Japan) with a Toshiba PVT-375BT 3.5 MHz transducer (Toshiba medical systems corporation, Otawara, Japan). Patients with mild acute pancreatitis were examined within 24 h of admission to the hospital. Color Doppler US was later repeated twice at three-month intervals. All US examinations were performed and reviewed by the same gastroenterologist specializing in the field. Participants were examined in the supine position during breath holding. The probe on the skin was applied and an insonation angle (the angle between the ultrasound beam and the blood flow) smaller than 60° was set in the middle part of the vessels (were the laminar velocity is the fastest) to measure the blood flow velocity, indices, and patency of the vessel. The main portal vein was assessed about 2 cm from the confluence of the splenic and superior mesenteric veins. Left and right portal veins were measured 1–2 cm from their branching. Each time, all abdominal organs were evaluated: liver, spleen, pancreas, as well as major hepatic vessels. Diameter, patency, blood flow, and main measurements and indices of portal veins were evaluated. The Doppler spectrum is a time–velocity waveform that represents variation in intravascular blood flow velocities during the cardiac cycle—systole and diastole [19]. Based on the spectral waveform of the vessel, mean indices were measured by the ultrasound machine:Vmax—maximum blood flow velocity—peak systolic blood flow velocity in the central layer of the vessel;Vmin—minimal velocity of blood flow—end diastolic blood flow velocity in the central layer of the vessel;Vmean—mean blood flow velocity in the central layer of the vessel;PI—pulsatility index—the parameter used to assess pulsatility defined as the difference between peak systolic blood flow velocity and minimum blood flow velocity, normalized to the time-averaged mean velocity;RI—resistivity index—the parameter used to reflect the resistance to blood flow caused by microvascular bed distal to the site of measurement. It is defined by the difference between peak systolic blood flow velocity and end diastolic velocity, normalized to the peak systolic blood flow velocity.

Vmax, Vmean, and RI were the measures that were used in statistical analysis and comparisons. The portal vein spectral analyses were always recorded three times to avoid influence of cardiac and respiratory fluctuations; the average was calculated for further statistical analysis.

Statistical analysis. Quantitative variables are expressed as the mean (± standard deviation) or median and range, and qualitative variables as absolute and relative frequencies. Comparisons between groups of quantitative variables were made by Wilcoxon–Mann–Whitney or Independent Sample t-test depending on their distributions. Chi square test was performed for qualitative variables. For paired samples, we used Cochran’s Q test after turning continuous data into dichotomous data. Pearson’s and Spearman’s correlation coefficients were used to determine possible associations. All tests were two-sided and *p* < 0.05 was considered significant. Data handling and analysis were performed with SPSS version 20 software (SPSS Inc., Chicago, IL, USA).

## 3. Results

Baseline characteristics of participants are presented in Table 1. Forty-five participants (32 (71%) males and 13 (29%) females) were included and completed this preliminary study. twenty-nine of them were diagnosed with mild AP, and 16 were healthy subjects. Both groups were homogeneous in terms of weight, body mass index, and age (*p* > 0.05). Male predominance was observed in the acute pancreatitis group (82.8%). Alcohol was the main etiological factor in the AP group (83% of all AP cases), followed by gallstones.

Heart rate and arterial blood pressure were measured in the morning of US examination. Mean heart rate of participants differed significantly between groups (*p* = 0.004), being on average five beats per minute faster in case of AP (Table 1). However, no difference was observed between AP and healthy individuals in terms of systolic and diastolic arterial blood pressure (131.0/82.4 ± 19.3/9.6 mmHg and 125.3/84.7 ± 9.7/7.4 mmHg, respectively). Heart rate and arterial systolic blood pressure did not correlate with maximal blood flow velocity (*p* = 0.719 and *p* = 0.466), mean blood velocity (*p* = 0.668 and *p* = 0.508), and RI (*p* = 0.684 and *p* = 0.649) in the main portal vein.

Results of blood tests showed significant differences between groups: an inflammatory pattern was observed in case of mild AP compared to normal values in the control group. Groups differed by the white blood cell count (11.2 ± 4.1 × 10^9^/L vs. 5.7 ± 0.8 × 10^9^/L respectively, *p* < 0.001) and the neutrophil count (77.2 ± 9.2% vs. 52.2 ± 7.4%, *p* < 0.001). Activity of p-amylase and lipase of AP patients was 24 and 85 times higher, respectively, compared to healthy individuals (*p* < 0.001) (Table 2). Activated partial thromboplastin time, prothrombin time, and international normalized ratio did not differ between groups and were within normal limits. However, fibrinogen and D-dimer were significantly higher in the case of AP: 6.1 ± 2.1 g/L vs. 2.8 ± 0.5 g/L; and 2482 ± 2416.9 µg/L vs. 136.6 ± 58.1 µg/L respectively; *p* < 0.001. Lactate dehydrogenase was also significantly elevated in the case of acute pancreatic inflammation: 263.6 ± 166.4 U/L compared to 168.5 ± 22.2 U/L in the control group (*p* < 0.001) (Table 2). Concentrations of electrolytes were significantly lower in the blood of AP patients; however, they remained within normal limits (Table 2).

Velocity of blood flow measured in portal veins did not differ significantly between the two groups during the acute phase of mild acute pancreatitis nor later, during the second or third follow-up visits (Table 3). All portal veins were completely patent during the follow-up visit in both study groups. Laminar blood flow was observed in all cases, which in the Doppler US spectra appeared as a broadening of the spectral line and filling of the spectral window.

Maximal blood flow velocity (Vmax) was statistically significantly similar when comparing results of healthy controls and acute pancreatitis patients: at the time of diagnosis, Vmax was 67.9 ± 29 cm/s in AP group compared to 67.5 ± 21 cm/s in control group (*p* > 0.05) (Figure 2). The same results were observed during the second and third examinations (Table 3). 

Vmax of the blood flow in the left and right portal vein branches did not differ between groups as well. It should be noted that for smaller vessels, values of maximal velocity of blood flow in the branches of the main portal vein were lower. Results of the Vmax of blood flow in the right portal vein branch were similar between the AP and control groups during the first, second, and third US examinations: 45.4 ± 27 cm/s vs. 44 ± 23.8 cm/s (*p* = 0.853); 41.7 ± 14.9 cm/s vs. 42.8 ± 26.6 cm/s (*p* = 0.559); and 42.3 ± 21.8 cm/s vs. 43 ± 22 cm/s (*p* = 0.797), respectively. Maximal blood velocity of the right portal vein branch was almost twice as fast as that of the left. However, as for other portal veins, maximal blood velocities in the left portal vein branch were not significantly different between AP and control groups (Table 4). Results between different examinations within the group did not differ either.

Mean blood flow velocity (Vmean) was similar in all three portal veins when comparing acute pancreatitis and control groups. Mild acute pancreatic inflammation had no significant effect on mean velocity of blood flow in the portal vein compared to healthy participants at the time of acute event (Figure 3), nor during second and third follow-up visits: 23.1 ± 8.5 cm/s vs. 24.5 ± 8.2 cm/s (*p* = 0.827); 22.3 ± 5.9 cm/s vs. 25.4 ± 4.5 cm/s (*p* = 0.317); and 24.2 ± 8.9 cm/s vs. 23.2 ± 7.2 cm/s (*p* = 0.883) respectively. In addition to this, Vmean of blood flow in the main portal vein did not change significantly during the postinflammatory six-month period (*p* > 0.05). The same results were observed in the right and left portal vein branches: Vmean of blood flow in those branches did not differ between study groups comparing all three examinations (Table 4); besides, this measure did not change during the study period.

Resistivity index (RI)—an automatically counted index—was the same in both study groups at the time of diagnosis in the main portal vein (*p* = 0.250) and in its right (*p* = 0.871) and left (*p* = 0.860) branches. As this index is counted based on the blood flow velocities, which did not change during the follow-up period, the RI of all three portal veins did not change significantly from the first to the last visits (Table 3).

Portal venous velocities did not correlate with demographic factors: gender, BMI, and age. It was not significantly associated with any of laboratory blood tests. Vmax of blood flow in the main portal vein did not correlate with age (*p* = 0.560; R = 0.070), BMI (*p* = 0.567; R = 0.069), white blood cell count (*p* = 0.698; R = 0.047), platelet count (*p* = 0.361; R = −0.110), activity of lipase (*p* = 0.125; R = 0.184), or *p*-amylase (*p* = 0.902; R = 0.016). This measure was not associated with coagulation parameters as well: it did not correlate with INR (*p* = 0.950; R = −0.008), APTT (*p* = 0.642; R = 0.058), fibrinogen (*p* = 0.259; R = 0.138), or D-dimer concentrations (*p* = 0.856; R = 0.022).

## 4. Discussion

Despite being a self-limiting disease with no signs and symptoms of local and systemic complications, mild acute pancreatitis was the target of this research. Mild AP comprises the majority of all AP cases; however, due to its benign course, not much attention is paid to it. Though both retrospective and prospective studies assessed the risk and incidence of portal vein thrombosis in the case of AP, no studies measured blood flow velocities in the portal venous system in mild acute pancreatitis. 

All AP patients included in this study were diagnosed with a mild AP based on the Revised Atlanta Classification. Patients presented with slightly elevated white blood cell and neutrophil count. Mean concentration of C-reactive protein—an important marker of the severity of acute inflammation—was 140.2 ± 94.8 mg/L and corresponded to the mild form of AP. Santorini consensus declares that a CRP value higher than 150 mg/dL is associated with severe AP [20]. Measured in the third day of the disease, CRP greater than 150 mg/dL had 80% sensitivity and 76% specificity, 67% positive predictive and 86% negative predictive value of severe AP. CRP greater than 180 mg/dL correlated with the formation of pancreatic necrosis, and sensitivity and specificity exceeded 80% [21]. As blood samples in the present study were collected within 24 h of the diagnosis, CRP reflected the acute inflammation and mild intensity of the disease (<150 mg/dL). In addition to this, CRP did not correlate with blood flow velocities in the portal system. 

Concentrations of D-dimer and fibrinogen are of significant importance as well—these parameters of the coagulation system have an effect and predictive value on the thrombosis in the portal system. In the present study, D-dimer value was nine times higher than the normal limit (2169.5 ± 2167.0 µg/L) and corresponded to the results obtained by other authors [16,22]. Results by Dumnicka et al. demonstrated that high concentrations of D-dimer were associated with more severe AP—severely (>10 times) increased levels were observed in 80% of severe AP patients compared to 10% of mild AP cases [22]. Furthermore, authors compared results of AP patients with and without DIC, and demonstrated that concentrations of D-dimer were twice as high in the case of coagulopathy (DIC), which is consistent with previous studies claiming that D-dimer concentration is associated with risk of venous thrombosis [23,24]. However, in the present study, all portal veins were completely patent at admission and during the follow-up period, and D-dimer concentration was not correlated with blood velocity in portal veins. Fibrinogen and other coagulation markers were within normal limits and this might be attributed to the mild course of the disease. Dumnicka et al., however, did not find any significant association between the severity of AP and fibrinogen concentrations, but it was significantly higher in case of DIC, suggesting its prognostic value for coagulopathy [22]. In addition to this, the authors found that low platelet count and an increase in prothrombin time and APTT were significantly related to the severity of AP. Low platelet counts, prolonged PT and APTT, and increased D-dimer concentrations were concluded to be suggestive of consumptive coagulopathy. The same was observed in a study by Kolber and colleagues [25]. These laboratory changes correlated with endothelial dysfunction (reflected by concentrations of angiopoetin-2 and soluble fms-like tyrosine kinase 1 (sFlt-1)) and urokinase-type plasminogen activator receptor (uPAR). The difference was that angiopoetin-2 and sFlt-1 correlated with the severe AP [22]; however, uPAR was not predictive of the severe AP at admission, but significantly predicted organ failure and ICU transfer or death [25]. However, the present study only included cases of mild AP; therefore, routine laboratory tests did not demonstrate signs of coagulopathy, nor disturbances in the blood flow on the US examination. 

In general, electrolyte abnormalities in AP are often associated with dehydration, prolonged vomiting, and calcium deposits in pancreatic fat. Serum potassium elevations may suggest hypovolemia and should be monitored and treated with fluid resuscitation and electrolyte replacement. Hypocalcemia is commonly found in AP patients requiring intensive care, especially in case of severe AP. Hypocalcemia is one of the components of Ranson’s scoring system for assessing the severity of pancreatitis, and is strongly associated with the severity of the AP and other complications [26,27]. However, in the present study, values of electrolytes were within normal limits in both study groups, being slightly lower in patients with AP. 

Blood flow is defined as the quantity of blood passing a given point in the circulation in a given period and can be expressed by the equation Q = VA, where Q is flow, V is velocity, and A is the cross sectional area of the vessel. Therefore, the velocity of blood flow is inversely proportional to vascular cross-sectional area. Since the vascular system obeys an adaptation of Darcy’s law, blood flow can also be expressed as the equation Q = ∆P/R, where ∆P is the pressure differential between the ends of the vessel and R is the resistance of the vessel wall. Resistance is a force that opposes the flow of a fluid, which in a vessel mostly depends on the vessel diameter. As vessel diameter decreases, the resistance increases and blood flow decreases. Measuring velocity only has its limitations, as it does not reflect the complete blood flow in the vessel. Blood flow, as well as velocity, is influenced by many factors, such as the cardiac output, compliance, volume, and viscosity of the blood, and length and diameter of the vessel. The Poiseuille equation describes blood flow and its relationship to known parameters [28]. The relationship between blood flow and pressure is exponential. An increase in arterial pressure not only increases the force that pushes blood through the vessels, but also distends the vessels at the same time, which decreases vascular resistance. Resistance is directly proportional to both fluid viscosity and the length of a vessel. Both of these affect the amount of friction—the greater either is, the greater the resistance and the smaller the flow. The radius of the vessel affects the resistance as well—the greater the radius, the greater the flow. Any change in the radius of a vessel has a very large effect on the resistance. As blood volume decreases, pressure and flow decrease. As blood volume increases, pressure and flow increase. Arterial blood pressure depends on cardiac output and total peripheral resistance. Increase in any of them increases arterial blood pressure. On the other hand, in the case of constant total peripheral resistance, an increase in arterial pressure increases blood flow and blood velocity. Blood pressure is related to peripheral resistance; an increase in peripheral resistance increases arterial pressure, but reduces organ blood flow and velocity of blood in vessels. In the present study, velocities of blood in the portal vein were measured, but cross-sectional area and length of the vessel were not. All veins examined with the color Doppler US were completely patent in the acute phase of the disease, as well as during the follow-up period. Blood flow was measured in the middle part of the vessel lumen, and laminar flow was observed in each case. Laminar blood flow refers to a distribution of flow velocities in layers that are parallel to the vessel wall. Near the vessel wall, where frictional forces are the greatest, the blood flow is the slowest; centrally, in the lumen of the vessel, frictional forces are the smallest; therefore, blood flow is the fastest. Results of the present study clearly demonstrate the association between the diameter of the vessel and blood flow velocity—both mean and maximal blood velocities were fastest in the portal vein, and decreased in the right and subsequently in the left portal veins as their diameters anatomically decrease in the same order: the main portal vein > right portal vein ranch > left portal vein branch. Perret and colleagues confirmed that the velocity of blood flow in vessels is closely related to cardiac output—an increase in cardiac output increases blood velocity in vessels and vice versa. Peak blood velocity was significantly correlated with systolic blood pressures between 135 and 160 mmHg; however, no correlation was found between peak blood velocity and blood pressures less than 135 mmHg or greater than 160 mmHg [29]. This corresponds with our results—blood pressure was not correlated with the velocities of blood flow in the portal veins and systolic blood pressure in each individual in both groups, and mean systolic arterial blood pressure was less than 135 mmHg in both groups. Blood viscosity has an effect on the blood velocity; however, in the present study, only patients with mild AP were examined, with no signs of shock or severe hypovolemia; in addition to this, liver disease was an exclusion criteria, meaning hypoalbuminemia was absent, and thus not affecting blood viscosity. Exclusion of patients with liver diseases and structural liver changes observed on US examination provided minimal differences in the compliance of vessels, which could have influenced the results of the measurements.

As it can be seen from the results, mild acute pancreatitis does not affect portal venous blood flow velocities; therefore, it is not associated with increased risk of portal vein thrombosis. We found no literature about the effect of mild acute pancreatic inflammation on the portal system’s blood flow, and apparently no studies were performed to test this. This can be attributed to the fact that this form of pancreatitis is not associated with local or systemic complications [9]. Some authors claim acute pancreatitis and pseudocysts in the case of chronic pancreatitis were the probable cause of spleno-portal vein obstruction in 91.4% of cases (half of them were related to pseudocysts of the caudal pancreas) [15]; however, others report that acute pancreatitis is more commonly associated with splenic vein thrombosis than portal vein thrombosis [30,31,32]. In addition, thrombotic complications are more likely to happen in the case of necrotizing, severe, or chronic pancreatitis [33,34,35,36]. This can explain why mild acute pancreatitis, in the absence of other prothrombotic factors (that were excluded based on the study protocol), has no negative effect on the blood flow in the portal system.

Despite its advantages in clinical practice, color Doppler US has its limitations that have been thoroughly discussed in medical literature: improper Doppler angle close to 90° may cause false positive velocities [37]; lower frequencies of US produce a deeper view but less accurate velocities [38]; the presence of turbulent flow may result in under or overestimation of the flow velocity [37,39]; respiratory and cardiac fluctuations can also affect measurements [40]. In our study, all 135 Doppler US examinations were performed by the same specialist gastroenterologist, thus allowing reliable comparison between data, as the same technique was used each time. According to Chuo et al.’s study of 48 healthy individuals, Vmean of the main portal vein (MPV) was 21.17 ± 8.15 cm/s and it did not change significantly between genders or different age groups [41]; Ulusan et al. measured MPV of healthy volunteers and Vmean was 32.15 cm/s [42]; Iranpour also claimed normal MPV Vmean should range from 20cm/s to 40cm/s [37]. This goes along with our results of healthy individuals—Vmean of the MPV was 24.5 cm/s in our study and it did not depend on the gender, age, or body mass index. Nevertheless, results of different authors have considerable variation, highlighting the need for studies of wider populations of healthy people. 

Brown et al. tested 35 volunteers and presented the results of normal portal venous flow, Vmean of the MPV was assessed to be 12.32 cm/s [40]; however, all other publications state that reduced blood flow and velocities dropping to less than 15 cm/s indicate portal hypertension and slowing of the blood flow in the portal venous bed increases the risk of thrombosis [15,37]. Such a decrease in mean portal venous blood velocity was not observed in our study in the acute pancreatitis group during the acute phase, nor during the follow-up period. In addition to this, portal venous velocities did not decrease significantly compared to healthy individuals; therefore, it can be claimed that mild acute pancreatitis does not affect portal venous flow. No publications exist on this issue; more are concentrated on already known pancreatic risk factors—chronic pancreatitis (CP) or severe acute pancreatitis. Bernades and colleagues arranged an eight-year follow-up study of 286 patients diagnosed with chronic pancreatitis. Velocities were not measured, but participants were examined for the thrombosis in the spleno-portal venous system. Only 5.6% (ten patients) experienced portal vein obstruction during the study time, and it was concluded that pseudocysts, especially of the caudal pancreas, and acute events of CP were the probable cause of spleno-portal venous obstruction in 91.4% of cases [15].
As it was mentioned earlier, portal vein thrombosis, as well as thrombosis in general, is a multifactorial condition caused by the interaction between three factors: inherited or acquired prothrombotic disorders, other thrombophilic factors, and local factors [8,11]. In this research, we aimed to take closer look at one part of Virchow’s triad—the blood flow in the portal system in mild AP—to find out if any changes occur during mild pancreatic inflammation that, together with trombophilic factors initiated by the inflammation, could trigger formation of thrombus. Recent studies have paid a lot of attention to the trombophilic state. Results of different studies on AP confirm the activation of the coagulation system leading to consumptive coagulopathy [16,22,25]. Even in the absence of clinically significant thrombotic complications, laboratory tests demonstrate the activation of coagulation and fibrinolysis. The degree of coagulation abnormalities in AP depends on the severity of inflammation—in the case of mild AP, thrombosis may be limited to pancreatic microcirculation, but it can be as severe as DIC in cases of severe AP. On the other hand, the association between coagulation and inflammation is two-sided: inflammation induces coagulation, but at the same time, activation of coagulation promotes inflammation [43]. The improvement in pancreatic blood flow inhibits the development of AP and accelerates the recovery. Despite this fact, the effect of early anticoagulant therapy has been argued in the literature. Different experimental studies have shown protective and therapeutic effects of heparin in AP: its inhibitive effect was observed on the development of AP induced by bile, taurocholate, or cerulein. However, heparin abolished the positive effect of ischemic preconditioning in the development of ischemia/reperfusion-induced AP [43,44]. Clinical studies also concluded that the positive effect of heparin in preventing AP after endoscopic retrograde cholangiopancreatography is insignificant [16]. Recent consecutive animal studies found a protective effect of acenocoumarol—a vitamin K antagonist—on the development of AP. Pretreatment with low doses of acenocoumarol prohibited the development of AP induced by cerulean or ischemia/reperfusion [43,45], and had a therapeutic effect in later stages [46].Limitations of our study are associated with the small number of participants and lack of moderately severe and severe cases of AP to compare the results with. Most cases of mild acute pancreatitis are treated in out-patient departments or in smaller peripheral hospitals; therefore, it would be useful to extend the study and include more medical centers, as well as moderately severe and severe AP cases, to increase the study population and obtain more reliable data and draw stronger correlations, and to compare the effects of different types of AP on the blood flow velocities of the portal system.

## 5. Conclusions

Portal blood flow velocities do not change during mild acute pancreatitis in the inflammatory and postinflammatory periods. This observation suggests that mild acute pancreatitis does not increase the risk of portal vein thrombosis. Portal venous velocities do not correlate with gender, age, BMI, coagulation factors, or enzyme concentrations in cases of mild acute pancreatitis.

## Figures and Tables

**Figure 1 medicina-55-00211-f001:**
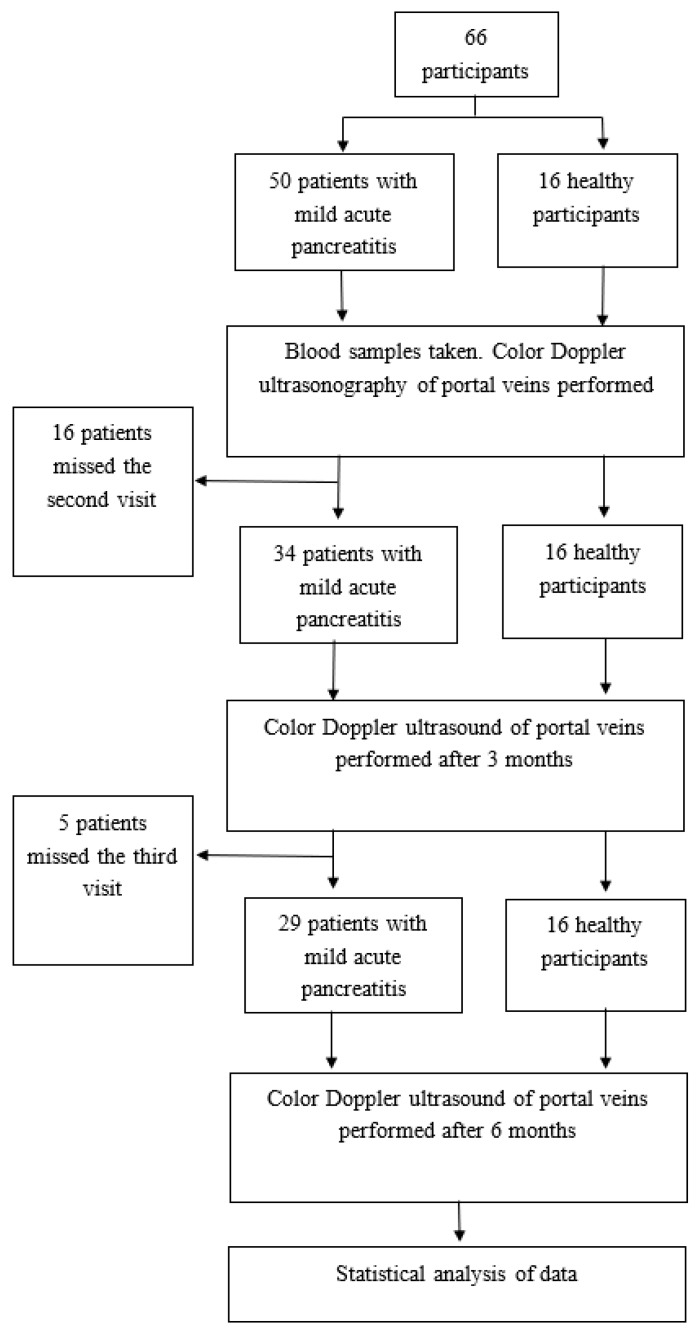
Flowchart of the study.

**Figure 2 medicina-55-00211-f002:**
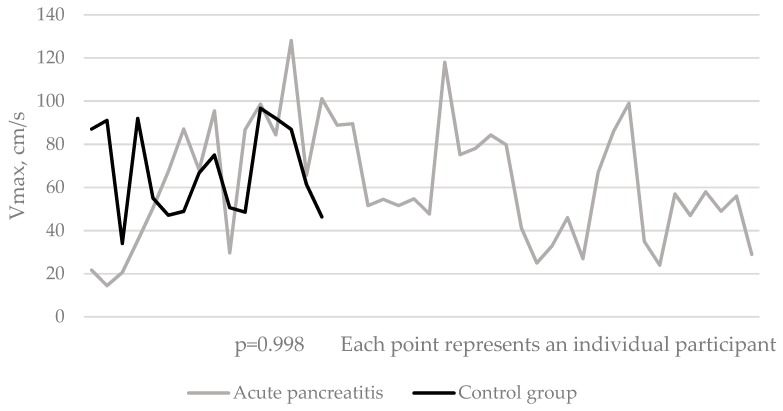
Distribution of the maximal blood flow velocity in the middle part of the main portal vein (where the laminar flow is the fastest) among study participants during the first examination.

**Figure 3 medicina-55-00211-f003:**
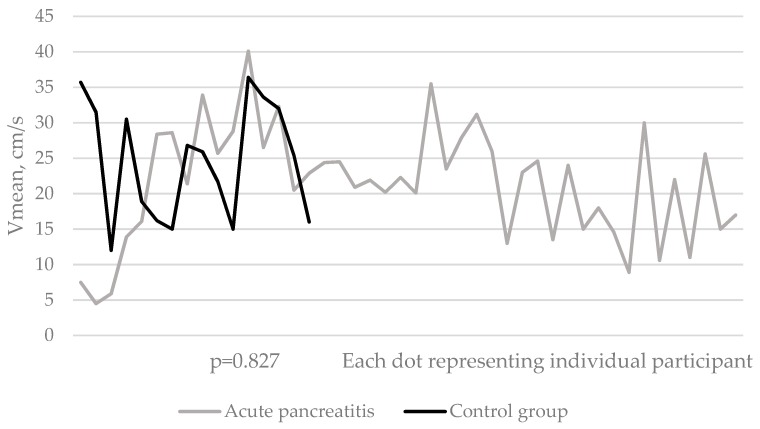
Distribution of the mean blood flow velocity in the middle part of the main portal vein (where the laminar flow is the fastest) among study participants during the first examination.

**Table 1 medicina-55-00211-t001:** Baseline Characteristics of Participants.

	AP Group *N* = 29 (64.4%)	Control Group *N* = 16 (35.6%)	*p* Value
Gender (%)			
Male	24 (82.8%)	8 (50%)	**0.037** *
Female	5 (17.2%)	8 (50%)	
Age, years	37.0 (23.0–69.0)	41.5 (25.0–63.0)	0.585 ***
Weight, kg	79.0 (59.0–133.0)	74.0 (56.0–102.0)	0.147 ***
Height, cm	179.0 (160.0–188.0)	175.5 (164.0–190.0)	0.845 **
BMI	24.6 (18.8–41.0)	24.3 (19.2–30.1)	0.112 ***
Heart rate, BPM	78.0 (60.0–110.0)	72.0 (64.0–78.0)	**0.004** ***
Systolic arterial BP, mmHg	130.0 (90.0–180.0)	127.5 (110.0–140.0)	0.337 ***
Diastolic arterial BP, mmHg	80.0 (70.0–100.0)	90.0 (70.0–95.0)	0.332 ***

Note: Data are presented as Median (Range); BMI—body mass index, BPM—beats per minute, BP—blood pressure, *p* < 0.05 expressed in bold. Measurements taken during the first examination. * Chi-square test, ** Independent Sample t-test, *** Wilcoxon–Mann–Whitney.

**Table 2 medicina-55-00211-t002:** Mean Values of Biochemical Test Results. Comparison between Groups.

Electrolytes	AP Group *N* = 29	Control Group *N* = 16	Normal Limits	*p* Value
Sodium, mmol/L	138.8 ± 3.2	141.3 ± 2.1	134–145	0.046 *
Potassium, mmol/L	4.2 ± 0.4	4.4 ± 0.5	3.8–5.3	0.041 *
Chloride, mmol/L	102.1 ± 4.9	104.1 ± 1.5	98–107	0.027 **
Calcium, mmol/L	2.1 ± 0.4	2.4 ± 0.1	2.1–2.55	0.033 **
**Biochemical markers**				
Lipase, U/L	3491.0 ± 5569.9	46.9 ± 57	8–78	<0.001 **
*p*-amylase, U/L	708.3 ± 1023.3	30.5 ± 21.1	8–53	<0.001 **
C-reactive protein, mg/L	140.2 ± 94.8	0.7 ± 0.7	<5	<0.001 **
Lactate dehydrogenase, U/L	263.6 ± 166.4	168.5 ± 22.2	235–243	<0.001 **

Notes: Data are presented as Mean ± Standard Deviation; * Independent Sample t-test; ** Wilcoxon–Mann–Whitney.

**Table 3 medicina-55-00211-t003:** Blood Flow Velocities in the Main Portal Vein in Patients with Mild Acute Pancreatitis and Healthy Volunteers during the Follow-up Period.

Parameter	AP Group	Control Group	*p* Value
Vmax 1st examination	67.9 ± 29.0	67.5 ± 21	0.998 *
Vmax 2nd examination	62.7 ± 19.2	65.7 ± 13.9	0.837 *
Vmax 3rd examination	68.2 ± 24.1	60.8 ± 20.6	0.496 **
Vmean 1st examination	23.1 ± 8.5	24.5 ± 8.2	0.827 *
Vmean 2nd examination	22.3 ± 5.9	25.4 ± 4.5	0.317 *
Vmean 3rd examination	24.2 ± 8.9	23.2 ± 7.2	0.883 **
RI 1st examination	0.4 ± 0.1	0.4 ± 0.1	0.250 *
RI 2nd examination	0.4 ± 0.2	0.4 ± 0.1	0.565 **
RI 3rd examination	0.4 ± 0.2	0.4 ± 0.2	0.929 **

Notes: Data are presented as Mean ± Standard Deviation. Measured in the center of the vein, where the velocity is greatest. * Independent Sample t-test, ** Wilcoxon–Mann–Whitney.

**Table 4 medicina-55-00211-t004:** Velocities of Blood Flow in Portal Veins in Patients with Mild Acute Pancreatitis and Healthy Volunteers during the First Examination.

Vein	AP Group	Control Group	*p* Value
Maximum velocity, cm/s			
Main portal vein	67.9 ± 29	67.5 ± 21	0.998 *
Right portal vein branch	45.4 ± 27	44 ± 23.8	0.853 **
Left portal vein branch	20.5 ± 10.9	21.2 ± 9.6	0.851 **
Mean velocity, cm/s			
Main portal vein	23.1 ± 8.5	24.5 ± 8.2	0.827 *
Right portal vein branch	16.4 ± 7.9	16.4 ± 8.1	1.000 *
Left portal vein branch	8 ± 3.4	7.4 ± 2.5	0.826 **
Resistivity index			
Main portal vein	0.4 ± 0.1	0.4 ± 0.1	0.250 *
Right portal vein branch	0.4 ± 0.2	0.4 ± 0.1	0.871 *
Left portal vein branch	0.4 ± 0.1	0.3 ± 0.2	0.860 *

Notes: Data are presented as Mean ± Standard Deviation. Blood flow velocity was measured in the center of the vein, where velocity is the greatest. * Independent Sample t-test, ** Wilcoxon–Mann–Whitney.

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
