# Peer review of "Changes in the Velocity of Blood in the Portal Vein in Mild Acute Pancreatitis—A Preliminary Clinical Study"

_medicina, 2019, doi:10.3390/medicina55050211_

Round 1
Reviewer 1 Report
Manuscript ID: medicina-434636
Title: Effect of Mild Acute Pancreatitis on the Portal Venous Velocities
Authors: Artautas Mickevičius *, Jonas Valantinas, Juozas Stanaitis, Tomas
Jucaitis, Laura Mašalaitė
The above manuscript is potentially interesting and undertakes a problem of the risk of portal vein thrombosis in patients with mild acute pancreatitis. However, numerous shortcomings and errors, lack of important information about of data, insufficient presentation of observation, over-interpretation of results, typing and language errors make the current version of the manuscript unacceptable for publication.
Major points:
1. The best method to evaluation of blood flow in vessels is measurement of volume flow expressed in units of volume per unit of time. However, color Doppler ultrasound does not allow to do that. Velocity of blood flow can be also useful. However, velocity of blood flow (V) in any vessel with linear flow depends on the volume flow (Q) expressed in units of volume per unit of time and the inner cross section area of the vessel. The relationship between them is in agreement with equation: V=Q/A.
For this reason, the authors should measure the cross section area of the portal vein and its branches, and obtained data must be presented in results and discussed. Moreover, the authors should present representative images of portal vein in both groups of participants. Lack of these data does not allow any conclusion.
Moreover, velocity of blood flow in vessels is related to cardiac output. An increase in cardiac output increases blood velocity in vessels; whereas a decrease in cardiac output decreases blood velocity in vessels. For this reason, the authors should evaluate the cardiac output by determine of stroke volume output (end-diastolic volume of ventricle minus end-systolic volume of ventricle) and heart rate. The reviewer understands that the study are already curried out. For this reason, the authors should at least present the heart rate and arterial blood pressure to assess the dynamics of circulation. Moreover, the authors should present factors affecting blood flow velocity in vessels and discuss the influence of these factors on their results.
2. In materials and methods, the authors have reported that patients with acute pancreatitis and healthy volunteers were examined three times with 3 months intervals. However, no data showing what was the time relationship between the onset of the disease and the time of the first examination. The time of the first examination should be standardized and clearly presented. The best idea was to perform the first examination at the day of hospitalization and repeat it at least at the 2nd, 3rd, 4th, 5th and 7th day of hospitalization, because the possibility of developing portal vein thrombosis is most likely in the early days of the disease. On the other hand, the authors are no longer able to carry out additional examinations in their study and for this they should present their research as a preliminary study, and show it in the title, abstract and discussion.
Minor points
1. The title of the manuscript. The authors did not measure the velocity of the portal vein but ever, the velocity of blood in the portal vein. Moreover, the methods and results exhibit some shortcomings (see major point 2). For this reason the title of the manuscript should be changed to “Effect of Mild Acute Pancreatitis on the Velocity of Blood in the Portal Vein. Clinical Preliminary Study”.
2. Page 1, line 6. “Biomedical science” change to “Biomedical Science” and add the city name.
3. Page 1, line 7. “hospital Santaros clinics” change to “Hospital Santaros Clinics” and add the city name.
4. Abstract the 1st sentence. It is not true. The portal vein thrombosis is just one of the reasons for the decrease blood flow velocity (see major point 1). For this reason this sentence should be changed to “The portal vein thrombosis is associated with a decrease in the main blood velocity in this vessel”.
5. Abstract, line 14. “…affect portal vein flow..” change to “…affects the portal vein blood flow”.
6. Abstract, line 17-18. The sentence “All participants were examined three times…” change to “All participants were examined three times. The first examination was carried out at the beginning of the disease, and the next two at three month intervals.”
7. Abstract, line 19-20. The sentence “Evaluated were mean and…” change to: “ Mean and maximal blood velocities and resistivity index in the main portal vein and its left, and right branches were evaluated.”
8. Abstract, line 20-21 The sentence: “Mean velocity of the main portal vein…” change to “Mean velocity of blood flow in the main portal vein…”. And the authors should change it in in other parts of the manuscript in the same way.
9. Abstract, line 27. The authors have not obtain sufficient evidences to support such a conclusion (see major point 2). The authors should change their conclusion to: “Mild acute pancreatitis had no effect on the portal blood flow velocity in the studied period of this inflammation and post-inflammatory periods. This observation suggests that mild acute pancreatitis does not increases the risk of portal vein thrombosis.
10. Page 1, line 41. All abbreviations should be presented in full form at the place where they are used for the first time.
11. Page 2 line 44-46. The sentence ”Researchers…” should be changed to ”Researchers have demonstrated a significant correlation between main blood flow volume and mean blood velocity in the portal vein, and the liver cirrhosis stage (9) (10); however, a decrease in the mean blood velocity in the main portal vein may be also a sign of portal vein thrombosis.
12. Page 2 line 65. Change “…by each of study participants.” to “…by each study participant.”.
13. Page 2, line 69. “…Hospital Santaros clinics” change to “Hospital Santaros Clinics”
14. Page 2 line 75. The authors should add some information about time of examination. They can use a sentences from abstract. Moreover they should present what was time relationship between the onset of the disease and the time of the first examination (see major point 2.
15. Page 2 line 76. The sentence “After the first visit…” should be changed to “After the first examination, 21 patients did not come to the second or third visit, therefore they were excluded from the further study (Figure 1).
16. Page 3, line 86. At the second sentence, the authors should write that “All patients and control subjects were asked to starve for at least 6 hours before blood sampling and US examination.
17. Page 4 line 94-96. The sentence “For the US examination, both…” should be removed.
18. Page 3, line 86-page 90. The authors should write same words about blood sampling and collection, as well as they should present in detail the methods and chemicals used for measurement od biochemical parameters tested.
19. Page 4, line 91. Before the word “first” should be written “the”.
20. Page 4, line 92. The authors should add name of city in the description about color Doppler ultrasound.
21. Page 4, line 103-108. The authors should present the definitions of mentioned parameters.
22. Page 4, line 113. “p<0,05” should be changed to “p<0.05”
23. Page 4, line 115-118. This paragraph should be removed because this information is already presented on page 2, line 64-67.
24. Page 4, line 120. “table 1” should be changed to “Table 1”.
25. Page 4 line 126-127, Table 1. The authors should also present mean ± SEM or SD, because this form allows for easier evaluation of statistical significance.
26. Page 5, line 135. Levels of electrolytes should be presented in detail in separate Figure or Table. Moreover results obtained during biochemical analysis should be discussed in Discussion.
27. Page 5 line 139-142. The authors should write what kind of blood flow they observed, laminar or turbulent. In the case of laminar blood flow, they should write about relationship between the layer of blood and blood velocity, and discussed it in Discussion.
28. Page 5, line 141; line 151, line 164, page 6 , line 183. Sentences should be corrected according to minor point 8.
29. Page 5, line 141. Change “(2 table)” to (Table 2).
30. Page 5, Figure 2. The authors should present the vertical, as well as the horizontal axis and the units on each axis. Figure legend: maximum should be changed to maximal, and the figure legend should describe what this figure presents. Probably, it is a representative record of the maximal blood flow velocity, in patients with pancreatitis and healthy volunteers, in the portal vein in the center of this vessel, where the velocity is greatest.
31. Page 5 line 168.. Change “3 picture” to “Figure 3”.
32. Page 6,.Figure 3. Correct this figure according to minor point 30.
33. 33. Page 6 line 185. Probably, the authors measured the activity of lipase and amylase, but not concentration.
34. Discussion, Conclusions should be changed according major point 1 and 2, minor point 9, 27
35. Page 6, line 193. Change “infection” to “inflammation”.
36. Page 7, line 199. Cange “What is more,…” to “In addition,…”.
Author Response
We appreciate the time and work You put in reviewing our paper. Such a detailed report allowed us to make the most precise corrections. We hope corrections we made answer Your questions and doubts about the study design and information presented in the manuscript. Thank You for the remarks on the study design, unfortunately the study is over and we cannot change the way and amount of data collected. The review You made is useful when conducting a more thorough clinical study on this subject. Correction made to the: Major points: 1. We added the information about heart rate and arterial blood pressure in the table titled Baseline characteristics. We put the paragraph about the influence of different factors on the blood velocities in general and on our results in the discussion section. 2. Blood sampling and color Doppler ultrasound were performed within 24 hours of hospitalization. We have put this information in detail in the section of Materials and methods twice – when talking about blood tests and US examination. This should now clearly demonstrate the relationship with the onset of the disease – that both US examination and blood sampling were performed on the first days of the disease, and that the time of examination was standardized. Minor points: 1. The title was changed according to the remarks 2. Corrected 3. Corrected 4. The sentence was changed according to the remarks 5. Corrected 6. Corrected 7. Corrected 8. Corrected here and in other parts of the manuscript 9. Corrected in the abstract and put the same way in the conclusion of the main manuscript 10. Abbreviation of Acute pancreatitis was presented with the first sentence of the introduction 11. Corrected 12. Corrected 13. Corrected 14. Corrected to the major point 2 15. Corrected 16. Corrected 17. Deleted 18. Corrected – we wrote a thorough explanation about the sampling procedure, time of specimen collection, analyzers used in the laboratory. 19. Corrected 20. City included 21. The definitions are now presented of all indices 22. Corrected 23. Removed 24. Corrected 25. Additional information included 26. We have put an additional Table presenting values of electrolytes and some biochemical parameters. Results of biochemical analysis are discussed in the Discussion 27. Information about the relationship between blood flow layer and vessel wall and blood flow velocities is included in the Discussion 28. corrected 29. Corrected 30. These Figures were completely removed and changed by two tables in respect with the remarks of another reviewer. Title of the Table and notes were corrected by Your remarks 31. Corrected 32. The same as with 30, Figure changed to Table 33. Corrected 34. Conclusions changed and supplemented with the remarks by other reviewers 35. Corrected 36. Corrected Thank You for the review
Reviewer 2 Report
There is little clinical and scientific significance to demonstrate that no portal venous thrombosis in mild acute pancreatitis. To make a complete study, moderate and severe pancreatitis should be included.
The presentation of portal venous velocities are difficult for readers to understand, particularly Figure 2 and 3. In addition the time after onset of pancreatitis is not clearly disclosed nor the diagnosis, for example the level of amylase, organ injury markers etc.
The manuscript is not well-written and need substantial editing to a publishable form
Author Response
Thank You for the remarks on our manuscript. We have corrected it a lot based on the remarks by three Reviewers We do understand a wider design of the research – i.e. inclusion of moderately severe and severe AP cases – would be more informative to make the significant conclusions about the effect acute pancreatitis has on the portal vein thrombosis. Unfortunately, the present study is already finished and no changes in the design can be made now. However, we changed the title of the manuscript according to the remarks by the Reviewer1 – “a preliminary study” to clarify the need for a further research and wider design of the study as well as limited conclusion. Figures representing portal venous velocities were removed and changed under Your remark to the tables presenting velocity differences between groups and between different examinations within separate groups. Hopes this makes it clearer. The Material and Methods section was supported by additional information: the relationship between the onset of the disease/time of diagnosis and time of the first examination; and blood sampling is now explained in detail. Biochemical markers, electrolytes and pancreatic enzymes are now presented in the table, to have a clearer view about the changes in case of mild AP. Thank You for the review
Reviewer 3 Report
According to revised Atlanta classification mild acute pancreatitis is self-limited disease not causing any local and systemic complications thus measurement of venous blood flow velocities seems not very clear to me. Nevertheless idea of measuring and describing venous flow changes in portal system during acute inflammatory process in the retroperitoneal space is rather innovative approach.
Data regarding the dynamics of inflammation is lacking (CRP, procalcitonin etc.). Also it would be very interesting to see overall hospital stay and treatment algorithm – use of intravenous fluids (volume, fluid type and duration of fluid therapy), use of anticoagulants, NSAIDS etc. Data regarding daily fluid balance is lacking, thus it can affect venous flow parameters.
There are only 45 patients included in study; numbers are very small for obtaining strong statistical evidence and drawing any strong conclusions. I cannot agree with authors that more patients with mild acute pancreatitis should be included to empower statistical results. Data comparing both groups already shows no statistical difference. Due to the fact that mild acute pancreatitis is self-limited disease, including more patients most probably will empower statistical results but will not show any clinically relevant differences between both groups.
It would be very interesting to compare mild acute pancreatitis with moderately severe or even severe acute pancreatitis. I strongly encourage authors for further research in more complicated forms of acute pancreatitis. This study can serve as a solid background for further research.
Author Response
We appreciate the time and effort You put in reviewing our manuscript. We have made a lot of changes based on the remarks received by three Reviewers. Yes, based on the Atlanta classification, as well as many other researches made, mild acute pancreatitis is both self-limiting, and causing no local or systemic complications. What is more, majority of patients are not hospitalized at all. However, we found several case reports of a self-limiting AP and acute portal vein thrombosis (all these patients had additional illnesses, however the onset of the acute pancreatitis was related to the onset of the thrombosis). In addition to this, other authors suggest repeated acute pancreatitis (not outlining the severity) in case of chronic pancreatitis are risk factors for the portal vein thrombosis. Therefore, it was a preliminary study to assess whether there are at least minor changes in portal venous velocities that could suggest the risk of portal vein thrombosis. According to the remark by the Reviewer1, we changed the title to pay attention to the fact that this study was a preliminary, and added this information in the Material and Methods section. According to Your remarks, we put additional information about the treatment algorithm. Biochemical parameters are now presented in a separate Table. Unfortunately, we did not register the dynamics of the inflammation – blood samples for the study were taken only on the first day of hospitalization (blood tests were repeated every day, or every two days in the in-patient department, however, registered for the study were only results of the first day test results). Conclusions were corrected according to Your remarks. Comparison between all types of AP were suggested by other Reviewer as well, therefore, it is in our future plans to design a study to compare portal venous velocities between mild, moderately severe, and severe AP patients, and make design changes based on remarks made by You and other Reviewers. Thank You for the review
Round 2
Reviewer 1 Report
Manuscript ID: medicina-434636
Title: Effect of Mild Acute Pancreatitis on the Portal Venous Velocities
Authors: Artautas Mickevičius *, Jonas Valantinas, Juozas Stanaitis, Tomas
Jucaitis, Laura Mašalaitė
The second review
The current version of the manuscript exhibits marked improvement of its quality. However, there are still some errors, which should be corrected.
Major points:
1. In former major point 1, the authors were asked for presenting factors affecting blood flow velocity in vessels and discuss the influence of these factors on their results. The have written only some words about relationship between arterial blood pressure and blood velocity. However, even this information is not complete. In Discussion, the authors should present limitations in blood flow assessment by measuring its velocity. They should also write that velocity of blood flow (V) in any vessel with linear flow depends on the volume flow (Q) expressed in units of volume per unit of time and the inner cross section area of the vessel., and the relationship between them is in agreement with equation: V=Q/A. It is important to present Poiseuille equation and discuss its impact on the blood flow velocity. Arterial blood pressure depends on cardiac output and total peripheral resistance. Increase in any of them increases arterial blood pressure. On the other hand, in the case of constant total peripheral resistance, an increase in arterial pressure increases blood flow and blood velocity. However, blood pressure is related to peripheral resistance; an increase in peripheral resistance increases arterial pressure, but reduces organ blood flow and velocity of blood in vessels. All these relationships should be presented in Discussion.
Minor points
1. A former minor point 21. The definition of Vmin and V mean is not clear. Vmin is a minimal velocity of blood flow measured in the central layer f the vessel, where velocity is highest or in the layer in contact with endothelin, where the velocity is lowest.
2. A former point 26. Results on electrolytes should be discussed.
3. A former point 27. The authors should also write in results what kind of blood flow they observed, laminar or turbulent.
4. A former point 30 and 31. Former figures 2 and 3 should be corrected, but not removed.
5. Abstract, Table 1, Table 2, Table 4 and the body of the manuscript. In English between whole numbers and decimal fractions should be a dot, not a comma.
6. Page 4, line 351-353 of the new version of the manuscript. The sentence ”The probe was applied at an insonation angle smaller than 60ﹾ in the middle part the vessels (where the laminar velocity is the fastest) to measure the blood flow velocity, indices and potency of the vessel.“. This sentence is unclear. Probably it should be changed to: “The probe was applied to the skin at a smaller than 60ﹾ so as obtain an ultrasound scan from the middle part of the vessels, where the laminar velocity is the fastest.
7. Page 7 line 756-757 of the new version of the manuscript. The sentence “Mild AP comprises majority of AP cases, however, due to its benign course is paid little attention to” should be changed to “Mild AP comprises the majority of AP cases, however, due to its benign course, not much attention is paid to it.”
8. Page 8, line863-865. The sentence ”Near the vessel wall, where frictional forces are greatest, the blood flow is the slowest; centrally with the lumen, blood flow is the fastest, therefore the sample volume was positioned in the center of vessels image in longitudinal section.”. This sentence is unclear. Probably, the authors have wanted to write: “Near the vessel wall, where frictional forces are the greatest, the blood flow is the slowest; centrally in the lumen of vessel, frictional forces are the smallest and therefore blood flow is the fastest. For that reason, we measured the velocity of blood in the center of the portal vein and its branches.”
9. Because the manuscript is almost ready for publication, the authors should perform some improvement of the introduction and discussion in their manuscript. There is complete lack of information on one of very important etiological factor for development of acute pancreatitis. The introduction and discussion should be completed with the following articles about pancreatic blood flow, pancreatitis and recent studies on coagulation activation (PMID: 18955758, PMID: 28208708, PMID: 28368336, PMID: 30262764, PMID: 23070084); Also, there is no comments on the latest research about effects of anticoagulants in AP (PMID: 28430136, PMID: 27754317, PMID: 26579579). These articles should be added to the manuscript and discussed.
Author Response
Thank You for additional remarks to make the paper better.
Major points:
1. In former major point àWe have put a paragraph about the physiology of the circulation and dependence of the blood flow on various factors, outlining the Pouisille’s equation.
Minor points
1. Blood flow velocities with ultrasound are always measured in the middle of the vessel where the flow is the fastest, but due to the systole and diastole, the waveform of the blood flow has highest peaks during the systole – Vmax, and lowest points during diastole (Vmin); in between them, the ultrasound machine calculates Vmean. We have made corrections to make the method clearer to the reader.
2. Results of electrolytes are discussed in paragraph in the Discussion section
3. Corrected
4. Figures were put back, explanations wrote in detail
5. Corrected in the tables and through all the text
6. It would be incorrect to write that probe was applied on the skin at an insonation angle smaller than 60° - the probe is applied on the skin at different degrees just to obtain the best view of the region/vessel, and then in the sample volume (that is in the range gate), the angle between ultrasound beam and the blood flow is manually corrected on the ultrasound machine to set the angle to a certain degree. For the sentence was unclear, we made some corrections to make it clearer
7. Corrected
8. Corrected
9. The introduction was supplemented with the information about two most common etiological factors for the development of AP as well as their pathophysiological explanation.
A paragraph about the prothrombotic factors and effect of anticoagulant therapy is included in the discussion section. Ant information regarding prognostic effect of laboratory tests on the severity of AP is included in the discussion

Reviewer 2 Report
No more comment
Author Response
Thank You
Reviewer 3 Report
Thank you for all changes made to this article.
I have only one comment regarding the fluid support. Please clarify the fluid therapy volume. You write “5 ml/kg Ringer's and 0.9% sodium chloride solution in the first day of hospitalization, reducing to 1.5 l per day at the end of hospitalisation”.
5 ml/kg is it during an hour?, per six hours?, for both solutions or only one?
Thank you in advance!
Author Response
Thank you for the remark, we have added the information regarding fluid therapy.
Round 3
Reviewer 1 Report
Manuscript ID: medicina-434636
Title: Effect of Mild Acute Pancreatitis on the Portal Venous Velocities
Authors: Artautas Mickevičius *, Jonas Valantinas, Juozas Stanaitis, Tomas
Jucaitis, Laura Mašalaitė
The third review
The current version of the manuscript is almost ready for publication. Only some minimal errors have been found.
The paper by Dumnicka et al. titled “The Interplay between Inflammation, Coagulation and Endothelial Injury in the Early Phase of Acute Pancreatitis: Clinical Implication” (Int. J Mol Sci 2017; 18(2). pii: E354) was listed two times in the References as a position 16 and 43. Moreover the year, volume and number of page should be expressed as following: 2017; 18(2). pii: E354.
The paper by Kolber et al. titled “Serum Urokinase-Type Plasminogen Activator Receptor Does Not Outperform C-Reactive Protein and Procalcitonin as an Early Marker of Severity of Acute Pancreatitis” (J Clin Med 2018; 7(10). pii: E305) was listed two times in the References as a position 25 and 44. Moreover the year, volume and number of page should be expressed as following: 2018; 7(10). pii: E305.
Also, in other papers published in Int J Mol Sci, the numbers of pages should be preceded by pii: E.
References. In actual position 22, Dumnicka et al the numbering of pages should be as following: Int J Mol Sci. 2017; 18(4). pii: E753. doi: 10.3390/ijms18040753.
References. Position 29. The internet connection should be deleted from names of authors and name of journal should be delayed.
All references should be shown in the same manner. Some of them show date of issue, some of them do not.
Author Response
Thank You for noticing.
